Adaptation, phylogeny, and covariance in milk macronutrient composition

Blomquist Gregory E. blomquistg@missouri.edu
Department of Anthropology, University of Missouri , Columbia , MO , United States of America
Gandini Patricia
Electronic publication date: 2019 Nov 13
Publication date: 2019
Volume: 7
Electronic Location ID: e8085
Received 2019 Jul 3; Accepted 2019 Oct 22
Copyright: ©2019 Blomquist
Copyright year: 2019
Copyright holder: Blomquist
License: This is an open access article distributed under the terms of the Creative Commons Attribution License, which permits unrestricted use, distribution, reproduction and adaptation in any medium and for any purpose provided that it is properly attributed. For attribution, the original author(s), title, publication source (PeerJ) and either DOI or URL of the article must be cited.
License URL: https://creativecommons.org/licenses/by/4.0/

Keywords: Phylogenetic comparative methods, Maternal energetics, Nutrition, Multivariate statistics, Parental care, Mammal, Life history

Funding: The author received no funding for this work.

==============================
Background

Milk is a complicated chemical mixture often studied through macronutrient concentrations of fat, protein, and sugar. There is a long-standing natural history tradition describing interspecific diversity in these concentrations. However, recent work has shown little influence of ecological or life history variables on them, aside from maternal diet effects, along with a strong phylogenetic signal.

Methods

I used multivariate phylogenetic comparative methods to revisit the ecological and life history correlates of milk macronutrient composition and elaborate on the nature of the phylogenetic signal using the phylogenetic mixed model. I also identified clades with distinctive milks through nonparametric tests (KSI) and PhylogeneticEM evolutionary modeling.

Results

In addition to the previously reported diet effects, I found increasingly aquatic mammals have milk that this is lower in sugar and higher in fat. Phylogenteic heritabilities for each concentration were high and phylogenetic correlations were moderate to strong indicating coevolution among the concentrations. Primates and pinnipeds had the most outstanding milks according to KSI and PhylogeneticEM, with perissodactyls and marsupials as other noteworthy clades with distinct selection regimes.

Discussion

Mammalian milks are diverse but often characteristic of certain higher taxa. This complicates identifying the ecological and life history correlates of milk composition using common phylogenetic comparative methods because those traits are also conservative and clade-specific. Novel methods, careful assessment of data quality and hypotheses, and a “phylogenetic natural history” perspective provide alternatives to these traditional tools.

Introduction

Patterns of animal parental care reflect the diversity of their life histories and adaptive solutions to ecological challenges (Clutton-Brock, 1991). Obligate provisioning of infants by adult females with mammary milk secretions is an ancient and unique aspect of all mammalian life histories. Milk is a complex mixture of chemicals with nutritional, immunological, and hormonal signaling functions (Power & Schulkin, 2016), which can change across different phases of maternal care (Langer, 2008).

Many hypotheses have been proposed to explain differences in milk composition among mammals (Ben Shaul, 1963; Oftedal & Iverson, 1995), but they often have limited taxonomic scope or explanatory power. Recently, a comprehensive analysis of all available high-quality milk macronutrient data (percentages of fat, protein, and sugar) identified a strong phylogenetic signal in milk composition and limited ecological and life history covariates (Skibiel et al., 2013). Additional details of the structure of this phylogenetic signal, such as early burst diversification versus Brownian motion, were not addressed (e.g. Harmon et al., 2010). Moreover, as in nearly all previous research on milk composition, each component was treated separately with univariate regression modeling. Both shortcomings are understandable given the rapid, recent development of multivariate phylogenetic comparative methods(Adams, 2014; Adams & Collyer, 2018) and techniques for describing phylogenetic signal (Hardy & Pavoine, 2012; Cornwell et al., 2014; Keck et al., 2016; Bastide et al., 2018).

Multivariate analysis of milk composition is desirable for several reasons. First, multivariate statistical methods are generally of higher power and can accurately account for correlations among the milk components with or without ecological predictors (Vargason et al., 2017). Second, there are strong arguments for a physiological and biochemical basis for coevolution among milk macronutrient concentrations. For example, the main milk sugar lactose draws water from blood into the mammary lumen resulting in higher volume but dilute milks (Shennan & Peaker, 2000). The fattiest milks, as seen seals, have little to no sugar and much less water than terrestrial mammals (Eisert, Oftedal & Barrell, 2013). Genetic correlations among macronutrient concentrations are also well described in dairy animals. For example, the correlation between protein and fat concentration is strongly positive (≈0.8; Analla et al., 1996; Othmane et al., 2002), mostly likely due to pleiotropic effects of alleles for genes that influence each trait. Third, macronutrient concentrations maybe be related through substitution to accomplish a similar nutritional goal. At least in some taxa, intraspecific and within-individual variation often shows compensatory shifts in fat versus sugar concentration such that the energy content remains stable (Power et al., 2008; Whittier et al., 2011).

Finally, many authors have recognized macronutrient compositions covary, such as the low-fat, high-sugar milks of most primates and perissodactyls versus the aforementioned high-fat, low-sugar milks of seals (Ben Shaul, 1963; Martin, 1984; Oftedal & Iverson, 1995; Hinde & Milligan, 2011). Moreover, the covariation and phylogenetic clustering were self-evident in visualizations either when simply decorating the tips of a phylogeny or in phylomorphospace plots (Sidlauskas, 2008) of the concentrations (Fig. 1). The later of these can be particularly illustrative when components were constrained to sum to 100% as in the right-angle mixture model of nutritional geometry (Raubenheimer, 2011).

Figure 1 Nutritional geometry of milk as phylomorphospace filling.

(A) Raw percentages (g/100 g) of protein and fat are plotted. (B) Percentages out of a sum of fat, protein, and sugar totaling to 100 are shown on the right. Dotted isoclines of sugar concentration are given every 25% in the with the highest percentage from sugar closest to the origin. Triangles are aquatic species in both panels.

There were two major goals of this paper. First, I further described the phylogenetic signal in milk macronutrient concentrations through univariate and multivariate statistics and visualizations. These were intended to describe the overall pattern of phylogenetic signal (e.g. Brownian motion v. early burst) and identify clades with quantitatively distinctive macronutrient concentrations. Second, I used multivariate phylogenetic regression to revisit the results of Skibiel et al. (2013) and tested for ecological predictors of milk composition while estimating the phylogenetic and residual covariance among macronutrients.

Materials & Methods

All data were initially taken from the supplementary material provided by Skibiel et al. (2013). I made a handful of alterations to the milk concentration database to ensure its quality. A simple check for quality is regression of dry matter concentration against the sum of fat and protein or fat, protein, and sugar (Oftedal & Iverson, 1995). Large outliers from this regression were inspected and fixed with values from the original publications (Myotis velifer dry matter, Arctocephalus gazella all measures) or removed altogether where it was also inconsistent (Thylogale billardierii, Perameles gunnii, Notomys cervinus, and N. mitchelli). For Leptonychotes weddellii, new data including a sensitive assay of sugar were available (Eisert, Oftedal & Barrell, 2013). For the three Papio species with differing ecological data but the same milk composition, I used only Papio anubis which is one of the two species from which the milk data were derived (Roberts, Cole & Coward, 1985) and is very similar ecologically to P. cynocephalus. Further augmentation of the database with more recent publications was not necessary to meet the goals identified above. I used an ordinal coding of aquatic adaptation to try and more sensitively capture this feature than the binary coding of Skibiel et al. Three species (Neovison vison, Castor fiber, Alces alces) were categorized as partly aquatic, three families as mostly aquatic (Ornithorhynchidae, Phocidae, Otariidae), and cetaceans were the only group categorized as completely aquatic.

For phylogenetic analyses, I used a set of 1000 mammalian trees (Faurby & Svenning, 2015). I matched names by hand between the dataset where nomenclature differed using GenBank’s taxonomy for preferred names. The trees were trimmed to the 124 species or subspecies in the dataset with the drop.tip() function from the geiger package (Pennell et al., 2014). After trimming only 209/1000 trees were unique indicating some phylogenetic uncertainty, but analysis was performed using a single consensus tree from the set computed with TreeAnnotator (Drummond et al., 2012). Continuous predictors were log10-transformed to reduce skew and centered by subtracting their means to ease interpretation of intercepts in regression models. Milk macronutrient concentrations were logit-transformed to accurately account for their [0-1] boundaries as proportions. While macronutrient concentrations have traditionally been analyzed on a log scale or untransformed (Ben Shaul, 1963; Martin, 1984), this approach is usually considered inappropriate when proportions are outside the 0.2–0.8 range (Warton & Hui, 2011; Schmid et al., 2013; Chen et al., 2017). All data manipulation and analysis were carried out in R (R Core Team, 2018). Alternatives such as beta regression or logistic generalized linear mixed models (GLMM) have not been adapted for the types of complex multivariate phylogenetic analysis desired here.

I used the MCMCglmm R package to explore a trivariate (fat, protein, sugar concentrations) phylogenetic mixed model (Hadfield, 2010). The suite of milk concentrations is predicted by the ecological variables used by Skibiel et al., while accounting for phylogenetic relatedness. This multivariate approach should be higher power to detect associations and it provides a phylogenetic variance–covariance matrix (VCV) that describes the coevolution of the concentrations under Brownian motion (BM). The variables include the aquatic habitat (noted above) and others as used by Skibiel et al.: arid habitat (binary), maternal body mass, adult diet coded categorically as carnivore, omnivore, herbivore, relative duration of lactation as the ratio of lactation length to the sum of gestation and lactation lengths, reproductive output as the ratio of litter mass and female mass, and an ordinal code for developmental stage at birth (precociality).

To identify clades with quantitatively distinctive milk composition I used the univariate, rank-based test introduced by Cornwell et al. (2014) and implemented in their R package ksi. This is an inherently phylogenetic test for clade distinctiveness that works by comparing the frequency distribution of a trait with and without a clade through a sample-sized scaled Kolmogorov–Smirnov test. The iterative algorithm identifies a series of clades of declining distinctiveness and reports ambiguity among neighboring nodes which may be the most recent common ancestor of the distinctive clade. For the milk composition data, I also adapted the package’s R code to allow for bivariate and trivariate versions of the test relying on the Peacock.test R package (Xiao, 2017). This allows testing clade distinctiveness of trait pairs (e.g., Fat-Protein) and the full set of all three concentrations (Fat-Protein-Sugar).

These rank-based tests were complemented with quantitative evolutionary modeling implemented in the PhylogeneticEM R package (Bastide et al., 2018). This is a multivariate phylogenetic method designed to detect instantaneous evolutionary shifts in sets of correlated traits, which may correspond to the invasion of new habitats or evolution of novel traits that characterize a clade. The method identifies clades that have different stabilizing selection optima (θs) and quantifies strength of selection (a common α) for the “pull” of selection toward those optima. It is similar in scope to other packages such as ℓ1ou and mvSLOUCH, but it uses a more complex model allowing for trait correlations (Bartoszek et al., 2012; Khabbazian et al., 2016). All R code for the analysis is provided as an Supplemental Information 1.

Results

Graphical exploration

Correlations between concentrations of fat, protein, and sugar were clear in bivariate plots (Fig. 1). The fat-protein relationship was simplest to display because there is less missing data than sugar. There was also obvious phylogenetic clustering of species when color-coded or when phylogenetic relationships were overlaid with inferred ancestral states in a phylomorphospace plot. Constraining the three percentages to sum to 100 produces a right-angle mixture model of nutritional geometry, where sugar concentrations are diagonal isoclines in the bivariate fat-protein plot. Unusual clades stood out in both displays. Most notable were primates, perissodactlys, elephants, and some bats with low fat, low protein, high sugar milks; pinnipeds with extraordinarily high fat, modest protein, and low sugar content; cetaceans with high fat and protein and low sugar; and most marsupials with modest fat, but high protein and sugar. Additional ecological variables also helped interpret the scatter. Nearly all the species classified as aquatic by Skibiel et al. had high fat and protein with low sugar concentrations.

Distinctive clades

Non-parametric statistical tests for the distinctiveness of these clades reinforced the graphical patterns. The top 5 clades were tabulated but typically the importance statistic drops off steeply from the highest-ranked clade (Table 1). In the univariate KSI tests, pinnipeds and cetaceans were noted for high fat while perissodactyls and primates were for low fat. The primate genus Eulemur was also identified for its further reduction in milk fat. For protein, primates, perissodactyls and Pteropus stood out for their low values. Ruminants or bovids + cervids were also flagged for protein which may reflect some ambiguity in testing the cetacean node. Primates and pinnipeds were the most distinctive clades for their opposing sugar concentrations, with a marsupial node, perissodactyls, and Pteropus also flagged for high sugar concentrations. Bivariate and trivariate tests generally corroborated these patterns.

Table 1 Distinctive clades from the Kolmogorov–Smirnov Importance (KSI) tests of Cornwell et al. (2014).

Well-known clades names are given with others left as number. See Fig. S1 for a phylogeny with nodes labeled.

Node	Rank	KSI	KSI/max	Nodesets	
Fat	
Pinnipeds	1	3.268	1.000	Pinnipeds; nd12; arctoids; nd10; carnivores; phocids	
Cetaceans	2	2.221	0.680	Cetaceans; whales	
Perissodactyls	3	2.160	0.661	Perissodactyls	
Primates	4	2.272	0.695	Primates; anthropoids; catarrhines	
Eulemur	5	1.677	0.513	Eulemur	
Protein	
Primates	1	3.150	1.000	Primates; anthropoids; catarrhines; cercopithecoids	
Perissodactyls	2	2.186	0.694	Perissodactyls; Equus; nd33	
Pteropus	3	2.178	0.691	Pteropus; bats; nd71	
Ruminants	4	1.906	0.605	Ruminants; bovids+cervids; bovids	
Eulemur	5	1.565	0.497	Eulemur	
Sugar	
Primates	1	3.301	1.000	Primates; anthropoids	
Pinnipeds	2	2.700	0.818	Pinnipeds; nd12; phocids	
nd116	3	2.169	0.657	nd116; diprotodonts; nd114; nd113; marsupials; nd117	
Perissodactyls	4	1.926	0.584	Perissodactyls; Equus	
Pteropus	5	1.947	0.590	Pteropus; bats; nd71	
Fat - Protein	
Pinnipeds	1	3.396	1.000	Pinnipeds; nd12; arctoids; nd10	
Primates	2	3.056	0.900	Primates; anthropoids; catarrhines; cercopithecoids	
Perissodactyls	3	2.480	0.730	Perissodactyls	
Cetaceans	4	2.237	0.659	Cetaceans	
Pteropus	5	2.162	0.637	Pteropus; bats	
Fat - Sugar	
Primates	1	3.433	1.000	Primates; anthropoids	
Pinnipeds	2	2.700	0.786	Pinnipeds; nd12	
Perissodactyls	3	2.497	0.727	Perissodactyls	
Bovids+cervids	4	2.390	0.696	Bovids+cervids	
nd114	5	2.141	0.624	nd114; nd113; marsupials; diprotodonts; nd116	
Protein - Sugar	
Primates	1	3.653	1.000	Primates; anthropoids	
Pinnipeds	2	2.700	0.739	Pinnipeds; nd12	
Marsupials	3	2.395	0.656	Marsupials; nd113; nd114	
Perissodactyls	4	2.276	0.623	Perissodactyls	
Pteropus	5	2.147	0.588	Pteropus; nd71	
Fat - Protein - Sugar	
Primates	1	3.697	1.000	Primates; anthropoids	
Bovids+cervids	2	2.846	0.770	Bovids+cervids	
nd12	3	2.698	0.730	nd12; pinnipeds	
Perissodactyls	4	2.447	0.662	Perissodactyls	
Marsupials	5	2.292	0.620	Marsupials; nd113; nd114	

Evolutionary modeling

Evolutionary modeling with PhylogeneticEM agreed with the distinctiveness of these clades and quantified the different selective optima for each. The best fitting number of selective regimes was K = 6 with K = 7 another very good alternative by the package’s penalized likelihood selection criterion (BGHml, Fig. 2). The regime shifts common to both solutions were pinnipeds, primates, perissodactyls, otariids, and the phocid genus Mirounga. The K = 6 and K = 7 solutions only differed in how they described selective regimes within marsupials: with K = 6 marsupials were placed within a common regime, while with K = 7 diprotodonts and the diprotodont species Setonix were placed in separate regimes. For either K = 6 or K = 7 multiple equivalent solutions were identified (degeneracy), but these only differed in the order of shifts within pinnipeds (see Supplemental Information 1).

Figure 2 PhylogeneticEM number of selection regime (K) penalized likelihood criteria.

K = 6 is the best, though K = 7 is a good alternative.

Selective optima (θs) were close to the average values seen within each clade (Fig. 3, Table 2). The clade optima for primates, marsupials, perissodactyls and pinnipeds were all deviations from the inferred root (12.01% fat, 7.62% protein, 3.56% sugar). Optima for otariids and Mirounga were additional deviations from the inferred pinniped optimum. The overall strength of selection or “pull” to these optima is weak (α = 0.07) which translates into a phylogenetic half-life of about 9.5 times the total height of the mammalian phylogeny. Thus, the regime shifts can be thought of as instantaneous jumps to novel values embedded in a process that is well approximated by Brownian motion.

Figure 3 Clade shifts identified by PhylogeneticEM with K = 6.

Clades with different selection regimes are marked with icons. Colored blocks for each concentration show departures from the root value for each species on the logit scale. Missing data for sugar concentration were imputed from the PhylogeneticEM model for 15 species. See Table 2 for clade names and selection optima (θ).

Table 2 PhylogeneticEM selection regime optima for K = 6 clades.

Values are back-transformed to raw percentages (g/100 g).

	Primates	Marsupials	Perissodactyls	Pinnipeds	Mirounga	otariids	
Fat	3.18224	6.83607	1.20587	59.78315	44.72456	43.04597	
Protein	2.36287	7.95330	2.37018	7.97800	8.36046	10.83272	
Sugar	7.18186	7.71032	6.06746	0.76505	0.00007	0.08824	

Phylogenetic mixed model

Multivariate phylogenetic mixed model prediction of milk composition from ecological and life history traits was largely consistent with results of Skibiel et al., despite removal of some taxa from their dataset, recoding of aquatic habitat, and use of a multivariate technique to incorporate correlations among the milk variables (Table 3). Increased carnivory resulted in a large significant increase in fat concentration, a small non-significant increase in protein concentration, and modest non-significant reduction in sugar concentration. Increased lactation length also caused a large reduction in fat concentration. However, novel patterns also emerged from the reanalysis. There was a nearly significant reduction of protein concentration in arid-adapted mammals, and a nearly significant increase with increasing reproductive output. Finally, increasing aquatic-adapted mammals had significantly reduced milk sugar concentration. There was a non-significant trend for increasing fat concentration with aquatic adaptation.

Table 3 Multivariate phylogenetic mixed model regression coefficients, credible interval and MCMC p-values.

Regression coefficients are for milk concentrations as logit-transformed proportions. Non-intercept P < 0.10 are in italics and P < 0.05 in bold.

	Fat	Protein	Sugar	
	β	CI	P	β	CI	P	β	CI	P	
Intercept	−2.161	(−3.945, −0.317)	0.024	−2.347	(−3.448, −1.252)	0.000	−3.204	(−4.599,−1.805)	0.000	
Arid (0/1)	−0.190	(−0.507, 0.141)	0.255	−0.166	(−0.355, 0.022)	0.083	−0.019	(−0.345, 0.310)	0.925	
Aquatic (ord.)	0.220	(−0.146, 0.621)	0.260	0.044	(−0.174, 0.268)	0.680	−0.376	(−0.734, −0.014)	0.049	
Diet, omnivore	0.194	(−0.197, 0.570)	0.324	−0.029	(−0.248, 0.191)	0.797	0.089	(−0.277, 0.484)	0.648	
Diet, carnivore	0.799	(0.226, 1.396)	0.007	0.162	(−0.173, 0.490)	0.326	−0.376	(−0.978, 0.289)	0.245	
Female mass	−0.049	(−0.262, 0.175)	0.652	−0.034	(−0.152, 0.090)	0.562	−0.103	(−0.298, 0.103)	0.313	
Rep. output	0.150	(−0.178, 0.519)	0.372	0.167	(0.001, 0.344)	0.053	−0.246	(−0.624, 0.136)	0.199	
Lactation length	−0.901	(−1.621,−0.220)	0.013	0.105	(−0.269, 0.479)	0.595	0.398	(−0.443, 1.227)	0.338	
Precociality (ord.)	0.066	(−0.127, 0.264)	0.500	−0.031	(−0.139, 0.088)	0.598	−0.039	(−0.234, 0.164)	0.703	

Phylogenetic heritabilities and correlations reaffirmed a strong phylogenetic signal in milk composition (Table 4). All of the phylogenetic heritabilities were very high (0.872, 0.976, 0.997). The phylogenetic correlations were all moderate to strong indicating coevolution of concentrations. The fat-protein correlation was moderate and positive (0.675), while fat-sugar was strongly negative (−0.750) and protein-sugar was moderately negative (−0.473). Residual correlations were weaker and all included zero within their credible intervals.

Table 4 Multivariate phylogenetic mixed model phylogenetic (upper triangle) and residual (lower triangle) correlations with 95% credible intervals.

Phylogenetic heritabilities are on the diagonal. Correlations and heritabilities excluding zero from their credible interval are in bold.

	Fat	CI	Protein	CI	Sugar	CI	
Fat	0.976	(0.934, 0.994)	0.675	(0.447, 0.815)	−0.750	(−0.920, −0.464)	
Protein	0.156	(−0.775, 0.688)	0.997	(0.980, 1.000)	−0.473	(−0.70, −0.119)	
Sugar	0.380	(−0.069, 0.994)	0.251	(−0.536, 0.898)	0.872	(0.680, 0.979)	

Discussion

I used current comparative methods to describe the phylogenetic signal and ecological correlates of milk macronutrient concentrations. The multivariate phylogenetic mixed model results recover those from the previous analysis by Skibiel et al. (2013), especially on the influence of diet and relative lactation length on milk fat and protein. However, I also found statistical support for increasingly aquatic mammals having milk that this is lower in sugar and higher in fat (cf. Oftedal & Iverson, 1995). Other non-significant trends for reduction of protein concentration in arid-adapted mammals and increased protein with higher reproductive output may be biologically meaningful. Finally, phylogenetic correlations indicate that all three concentrations have coevolved during mammalian evolution history. Differences from the previous report could be due to different coding of predictors (aquatic), more stringent data filtering, and the Bayesian multivariate framework adopted here.

In general, there were few ecological and life history correlates of milk macronutrient composition with detectable statistical associations by these phylogenetic comparative methods. This contrasts with long-standing characterizations of milks as finely attuned to the reproductive ecology and maternal energetics of different mammalian clades (Oftedal & Iverson, 1995; Ben Shaul, 1963). The other tools implemented here were more consistent with these characterizations. Clades with distinctive milks (especially primates and pinnipeds) were routinely identified with rank-based tests. This was corroborated with statistical models that identified shifting selection regimes for each clade.

The disconnect between traditional phylogenetic comparative methods (PCMs) as regressions that “control for phylogeny” (Freckleton, Harvey & Pagel, 2002) with these distinctive clade and selection regime identification methods is striking. Because mammalian life histories and ecology are fairly conservative, there are few independent cases of mammals evolving similar ecological or life history traits that PCMs rely on to identify associations. For example, true aquatic adaption is only found in cetaceans, pinnipeds, and the platypus. PCMs are unlikely to identify statistical associations in this case. In contrast, diet categories are more diverse within mammalian clades such that a robust diet-milk fat association can be found by PCMs. While the success of traditional PCMs capturing this association may be due to a strong causal biological mechanism, failure to capture other hypothesized associations with precociality, or aquatic or arid habitats should not be used as evidence against these ecological factors influencing milk composition. Indeed, their authors advocate these newer methods as a “natural history tool” that effectively complement graphical and other descriptive methods (Uyeda, Zenil-Ferguson & Pennell, 2018). In particular, they are sensitive to clade-wide adaptations and novel lineage-specific traits that traditional PCMs fail to capture.

The most dramatic of these contrasts in clades was between primates and pinnipeds. These groups are well-known to have strikingly different strategies of infant provisioning and growth rates (Power & Schulkin, 2016; Skibiel et al., 2013; Hinde & Milligan, 2011; Langer, 2008; Oftedal & Iverson, 1995; Martin, 1984; Ben Shaul, 1963). Anthropoid primates in particular have the lowest average growth rates among placental mammals and feed infants “on demand” over a prolonged period of lactation where mother and infant are in close proximity allowing for frequent nursing (Charnov & Berrigan, 1993; Case, 1978). Pinnipeds grow much more rapidly over shorter lactation periods. Rapid energy transfer to infants is likely facilitated by maternal nutrient stores in blubber and, in some habitats, strongly selected for by the thermoregulatory demands of cold ocean water. Phocid mothers fast during very brief lactation periods while otarrid mothers forage and nurse infants during rare visits over longer lactation periods that are short for their body sizes (Sapriza, 2019; Schulz & Bowen, 2005). In both pinniped families, milk sugars other than lactose predominate. This allows for their milk to have very little water because lactose draws water into the mammary lumen (Shennan & Peaker, 2000).

Perissodactyl milks are similar to those of anthropoid primates. While they share “on demand” feeding with primates, the high lactose and thus water content of equid and rhino milks are argued to be necessary for evaporative cooling of infants in hot environments (Hinde & Milligan, 2011; Oftedal & Iverson, 1995). Traditional PCMs are ill-suited for handling these convergent milk phenotypes for the different ecological and life history correlates in these clades.

Marsupial milks are distinctive for high sugar concentrations coupled with moderate protein and fat concentrations. Such high sugar concentrations are accomplished by having very little lactose which allows marsupial milks to avoid becoming extremely dilute. In general, marsupials grow at a slow average rate over a long period of lactation (g/day, Case, 1978), which may allow brain growth to be dissociated from metabolic rate in marsupials (Weisbecker & Goswami, 2010). The clade shift identified here for marsupials is consistent with many recent studies arguing marsupial development is derived compared to the last common ancestor of marsupials and placentals (e.g., Werneburg et al., 2016). Comparative milk databases have used composition at pouch emergence (or teat detachment). The comparability of this stage to mid-lactation in placentals is uncertain as another distinctive feature of marsupial milks is their pronounced change in composition over the lactation period (Oftedal & Iverson, 1995).

The comparative database of milk macronutrient composition, while the best resource available, was inevitably limited. Standardization of data collected across taxa is not always clear and intraspecific variation is not always documented. This will appear as “measurement error” in comparative analysis and will reduce both phylogenetic signal and the strength of regression coefficients in PCMs or other analyses (Silvestro et al., 2015; Hardy & Pavoine, 2012). For example, prior to eliminating some of the concerning data points in the original database, these outlier species were often assigned there own selection regime. This is likely for elephant seals (Mirounga) assignment to their own selection regime. The milk sugar concentrations reported for these two species are very low and were treated as inequalities below detectable levels in original publications. Moreover, while the database is also adequate for many analyses its size will limit the power of recent statistical models designed to measure phylogenetic signal or discriminate among different patterns of selection (e.g., stabilizing, early burst) versus Brownian motion (Housworth, Martins & Lynch, 2004; Boettiger, Coop & Ralph, 2012; Silvestro et al., 2015; Uyeda & Harmon, 2014).

Conclusions

Mammalian milks are diverse but often characteristic of certain higher taxa. This makes the ecological and life history correlates of milk composition difficult to identify using traditional phylogenetic comparative methods because those traits are often conservative and clade-specific, too. Primates and pinnipeds have the most outstanding milks according to multiple newly devised tests, with perissodactyls and marsupials as other interesting clades.

Supplemental Information

Supplemental Information 1 Raw data, R code, and complete results of milk composition analysis

Click here for additional data file.

Thanks to Steve Leigh for an introduction to phylogenetic comparative methods and to Katie Hinde and Lauren Newmark for valuable discussions of lactation biology. Comments from two reviewers and the associate editor improved this manuscript.

Additional Information and Declarations

Competing Interests

Author Contributions

Data Availability

The author declares there are no competing interests.

Gregory E. Blomquist conceived and designed the experiments, performed the experiments, analyzed the data, contributed reagents/materials/analysis tools, prepared figures and/or tables, authored or reviewed drafts of the paper, approved the final draft.

The following information was supplied regarding data availability:

Raw data and code are available in the Supplementary Files.

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
