# Peer review of "Adaptation, phylogeny, and covariance in milk macronutrient composition"

_PeerJ, doi:10.7717/peerj.8085_

## Round 0.1 · original submission · Major Revisions

The study is well presented, I feel that the manuscript is dealing with a good topic but lacks in the quality of preparation. The main problem found in the manuscript is related to the some aspects of the methodology and technical corrections. Please use the past tense on the text, avoid singular sentences. It is necessary to improve the manuscript by examining the questions that need to be clarified in a way. Please be aware of the manuscript should be presented according to guidelines for authors of PeerJ. For your guidance, you can check the reviewers' comments. Thank you for your efforts. Regards,

·

Basic reporting

This is a well done study though some more details are needed in the methods section.


Please see the attached PDF for minor questions/edits related to grammar/sentence structure.

Experimental design

Line 69: It would be helpful to include the total number of species in the analysis (written in the text and not just the supplement)..
Line 101: It would be helpful to include the R code in the supplement so the reader could better evaluate the models.
102-103: Please explicitly list the predictor variables and why they may be important for milk composition.
Line 107: Please explain this analysis in more detail? Also, if this is not a phylogenetic test, then why use it?
Line 110: Explaining this method a bit more would be good.

Validity of the findings

Line 139: You mention alternative models are similar but it is not clear how this was judged? Is it based on AIC?

Line 169:170: So doesn't this support the idea that using more traditional or recently developed phylogenetic methods have little influence on the results? i.e. recently developed PCMs are not advantageous (at least with this dataset)

Line 191: Though it may also be possible that diet has a stronger association with milk composition due to biological factors

Additional comments

Some additional comments about the figures and tables:
Fig2A is too small. Fig2b&C is also too small and should be full page width.
Fig3 is quite small. The tree is not labeled so it's difficult to discern each clade's identity
Table 3: can you highlight predictors with p values <0.1 and <0.05

Reviewer 2 ·

Basic reporting

The manuscript is well written. The discussion needs greater development. Several of the figures are hard to read and need additional labels for clarity.

Experimental design

This paper is original and appropriate for PeerJ. The questions are well defined and while the results have some overlap with prior work, I think it is critically important that we revisit taxon-wide studies evaluating important adaptations, such as milk composition, as new methods become available. The methods appear to be in sufficient detail, however, I am not an expert on multivariate phylogenetic comparative methods.

Validity of the findings

The authors did not note if the data are available. However, the data from Skibiel et al are available at J Anim Ecol. The data are sound and appropriate measures have been taken to ensure that the data are valid. As described below, the discussion focused more on the value of the method that the findings themselves. I would have like to hear a lot more about the significance of the data. The author explicitly presents the result as selection regimes - this will be exciting to several readers. Thus, the patterns observed warrant further discussion.

Additional comments

I have reviewed Blomquist’s manuscript titled ‘Adaptation, phylogeny, and covariance in milk macronutrient composition (#34219). This study updates the findings of Skibiel et al. (2013) by applying multivariate phylogenetic comparative methods to the data. The results of this analysis confirm many of the findings of the original study and add the findings that primates are unique and suggest that distinct selection regimes underlie the evolution of milk in pinnipeds, primates, perissodactyls, and marsupials. While the results of this study have some overlap with Skibiel et al., I think it is critically important that we revisit taxon-wide studies evaluating important adaptations, such as milk composition, as new methods become available.

Major comments. Overall, I think the author did a nice job with this reevaluation. I particularly appreciated the nutritional geometry and the clade shift plots (but see comments below). A lot of the data presented will be unclear to a reader who is not familiar with multivariate phylogenetic comparative methods. Further, the phylogenies are small and not labeled, and thus, lactation biologist interested in these patterns will have a hard time gleaning patterns upon inspection. The discussion focused more on the value of the method that the findings themselves. I would have like to hear a lot more about the significance of the data. The author explicitly presents the result as selection regimes - this will be exciting to several readers. Thus, the patterns observed warrant further discussion.

Specific comments.
• Perissodactyl is misspelled throughout the manuscript.
• Line 18: replace ‘less sugary and more fatty milks’ with more from language ‘milk that this is lower in sugar and high in fat’
• Line 120: replace ‘Unusual clades stand out both displays’ with ‘Unusual clades stand out in both displays’
• Line 128-129: regarding the statement: ‘The top 5 clades are indicated but typically the distinctiveness drops of steeply from the highest-ranked clade’, please clarify where the ranking is coming from.
• Figure 2: A - I can’t read the titles for the axis. Further, please give a clear explanation of what A represents in the figure description. For B & C - it is unclear how the three trees differ. What are the numbers? They are very hard to read. Please explicitly give the color associated with each taxon.
• Line 154: change ‘resulted a large significant increase’ to ‘resulted in a large significant increase’
• Figure 3: please name the taxa associated with each color.
• Table 3: I would be much easier to interpret this is if ‘regression coefficient, (credible interval, MCMC) p-value’ were placed above each column. I recommend making all significant p-values bold to help your reader pick them out.
• Table 4: see Figure 3 – please make it more explicate what is in each column.
• Line 172: see comment for line 18

---

## Round 0.2 · accepted · Accept

Thank you very much for making the changes proposed by the reviewers. Your paper is ready for publication in PeerJ!

·

Basic reporting

The author has done a good job revising the figures and revising the text

Experimental design

I appreciate the added methodological details, including the additional code in the supplement.

Validity of the findings

This is a rigorously performed study

Additional comments

I appreciate the edits made to this latest version. I look forward to seeing the paper published.